# RecipeIS—Recipe Recommendation System Based on Recognition of Food Ingredients

**Miguel Simões Rodrigues** [1,*]**, Filipe Fidalgo** [1,2] **and Ângela Oliveira** [1,2]

1 Polytechnic Institute of Castelo Branco, 6000-767 Castelo Branco, Portugal
2 R&D Unit in Digital Services, Applications and Content, Polytechnic Institute of Castelo Branco, 6000-767 Castelo Branco, Portugal; ffidalgo@ipcb.pt (F.F.); angelaoliveira@ipcb.pt (Â.O.)
* Correspondence: msrodrigues105@gmail.com

**Abstract:** Currently, food waste is a global concern, a problem that arises mainly at the consumption level and generates environmental, economic, and social impacts. One way to reduce the food waste problem is to use the food we already have at home. However, this causes another concern, which is what to cook with certain foods. Sometimes we do not know what recipes can be made. Knowing which ingredients can be mixed and how to mix them can be a difficult task for a beginner cook, so selecting the right ingredients for a recipe is essential. Therefore, it is proposed to develop a recipe recommendation system through image recognition of food ingredients. Presently, the system is a web application that recognizes an image given by the user and recommends recipes containing the recognized ingredient. For this, a convolutional neural network model, the ResNet-50, was built to perform image recognition and trained with a dataset that contains about 36 classes of vegetables and fruits. Through this training, the model reached 96% accuracy in classifying the dataset images. The recommendation system uses the label of the recognized ingredient to obtain the recipes, which are searched through the Edamam API.

**Keywords:** food ingredient recognition; artificial neural network; dataset; ResNet-50; recommendation system

## 1. Introduction

Currently, it is possible to find several websites or mobile applications that have user profiles for their use. These websites or applications use a recommendation system that is based on the information given by users in their initial registration, such as preferences or tastes. In the culinary industry, there are several apps and websites that are used to find recipes based on keywords such as ingredient name, type of cuisine, or culture. These applications attend to the needs and interests of their users but fail to identify restrictions, such as the maximum limit on the number of ingredients. In such cases, users end up missing ingredients or replacing them with similar ones. To help users avoid such adjustments, ingredient recognition can be used to identify the ingredients that the user has at his or her disposal at that moment and, through that ingredient or those ingredients, recommend various recipes. The main objective of the recipe recommendation system through the identification of ingredients is to help users who wish to cook with the ingredients that they have available at home avoid food waste. This system can be used not only at home but also in a supermarket, using the smartphone to identify the ingredients. That way, the user can build a plan or idea of what to cook for the week based on the recommendations from the system. For food image recognition or object recognition, deep learning models and libraries like TensorFlow [1], Keras [2], and Numpy [3] are used. With these libraries, it is possible to implement pre-trained deep learning models such as ResNet, MobileNet, and VGG16, among others. Next, the use of a dataset is necessary to train the deep learning model so that it can learn from the input images given by the dataset. This

way, the model will recognize patterns in the images that will allow it to classify the image in a final stage.

Hence, it is proposed to use the model ResNet-50 to solve the problem of the recognition of food ingredients. Furthermore, a dataset with ingredient images is applied, which will give the model input images. Then, some convolution layers are added in the final part of the ResNet-50 model architecture to increase its accuracy level. With these changes, we obtained a value of 96% accuracy in the classification of food ingredients.

By incorporating ResNet-50, we were able to effectively analyze and classify ingredients based on their visual characteristics. This empowered us to provide accurate and reliable recommendations, as the model excelled at identifying ingredients with good precision. Moreover, our project seamlessly integrated with the Edamam API database, a comprehensive and extensive repository of culinary information. By harnessing the capabilities of this API, we gained access to a vast collection of recipes, nutritional data, and cooking techniques, eliminating the need for new database development.

To provide a comprehensive understanding of the project's structure and progression, a flowchart shown in Figure 1 was created to illustrate the tasks of this work. This visual representation identifies the stages of Project Planning, Analysis, Development, and Implementation.

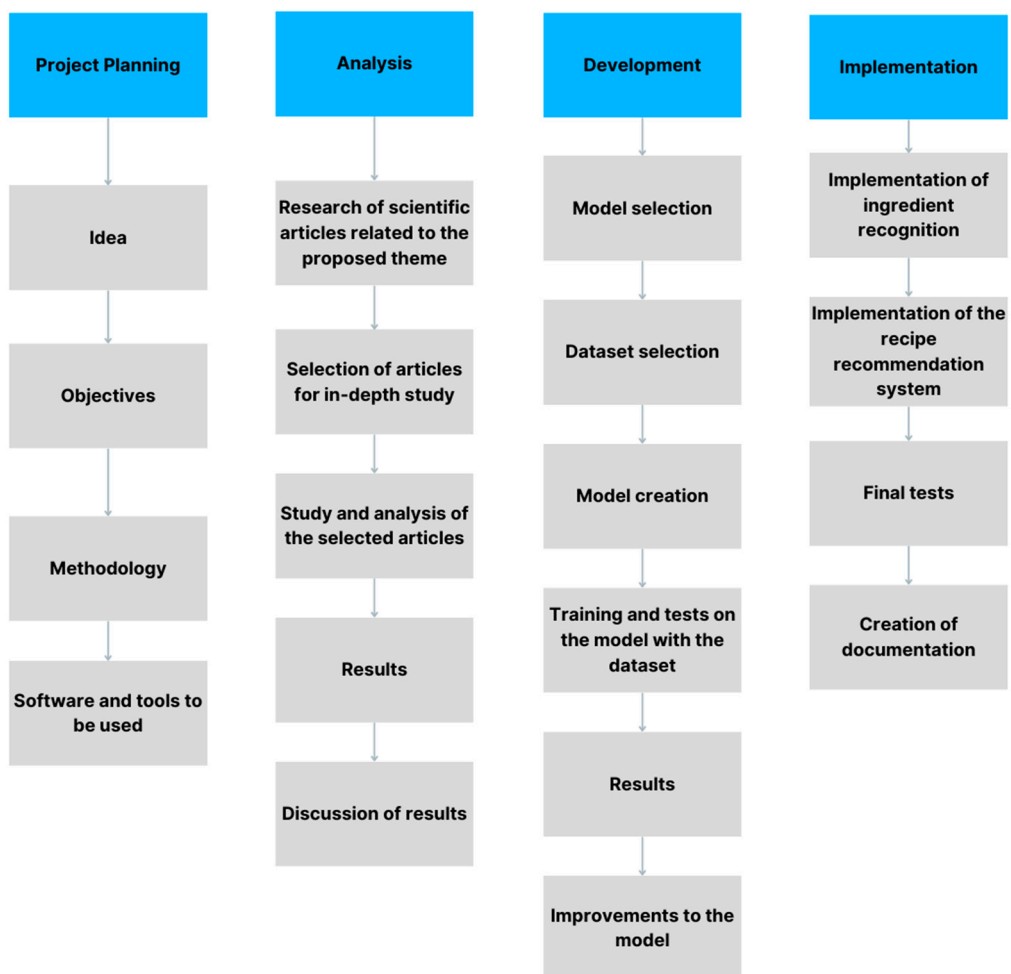

**Figure 1.** Flowchart of the project phases.

The article is structured as follows: Section 2 presents the research method supported by systematic revision methodology. Section 3 presents the materials and methods used in the development of the project, presenting the proposed Machine Learning model and its architecture, the dataset used, and then the training of the model together with the dataset.

In Section 4, the complete implementation of the project is presented, subdivided into two parts: the first is the implementation of ingredient recognition descriptions, and the second is the recommendation of recipes based on the recognized ingredient implementation. Finally, in Section 5, the conclusions and suggestions for future work implementation are shared.

## 2. Related Works

Recognition of food, ingredients, and recipe recommendations has gained popularity with time. Machine learning methods utilize artificial convolutional neural networks to recognize and classify patterns in images. Through research, several articles were found that discuss food ingredient recognition and recipe recommendation. This systematic review follows the PRISMA methodology (Preferred Reporting Items for Systematic Reviews and Meta-Analyses) [4]. The following topics were included:

A.    Research questions
B.    Inclusion criteria
C.    Research strategy
D.    Results
E.    Data extraction and analysis
F.    Discussion

To conduct the research, we formulated research questions on the topics of food image recognition and recipe recommendation. These questions aim to propose solutions to the identified problem. The research questions we used are:

- Question 1: Are there digital solutions that offer recipe proposals based on ingredient recognition or ingredient images?
- Question 2: What are the frameworks for ingredient recognition solutions?
- Question 3: Are there digital solutions for recipe proposals based on leftover meals to fight food waste?

Inclusion criteria refer to the key characteristics of the target population that researchers will use to answer the study question. The inclusion criteria defined for our study are as follows:

- Criteria 1: Studies from 2014 to 2021;
- Criteria 2: Studies written in English;
- Criteria 3: Studies proposing recipes based on ingredients or on the recognition of ingredient images;
- Criteria 4: Studies about image recognition of food ingredients;
- Criteria 5: Studies where the full text is available.

We searched for relevant articles using the IEEE Xplore [5], ACM Digital [6], and ScienceDirect [7] databases. The search terms we used included "ingredient recognition" AND "application", "package" AND "packaged food" AND "food recognition", "frameworks" AND "recipe recommendation", and "food waste recipes" AND "application". We searched between November and December 2021.

After applying criterion 1, we found a total of 146 scientific studies, with 85 from IEEE Xplore, 46 from ACM Digital, and 15 from ScienceDirect, as presented in Figure 2. We then applied criterion 2 and removed duplicates, resulting in 130 studies. We analyzed these studies based on criteria 3 and 4. We then performed a full-text analysis of the remaining 79 studies, applying criteria 3, 4, and 5. Based on these criteria, we included 16 studies in our review.

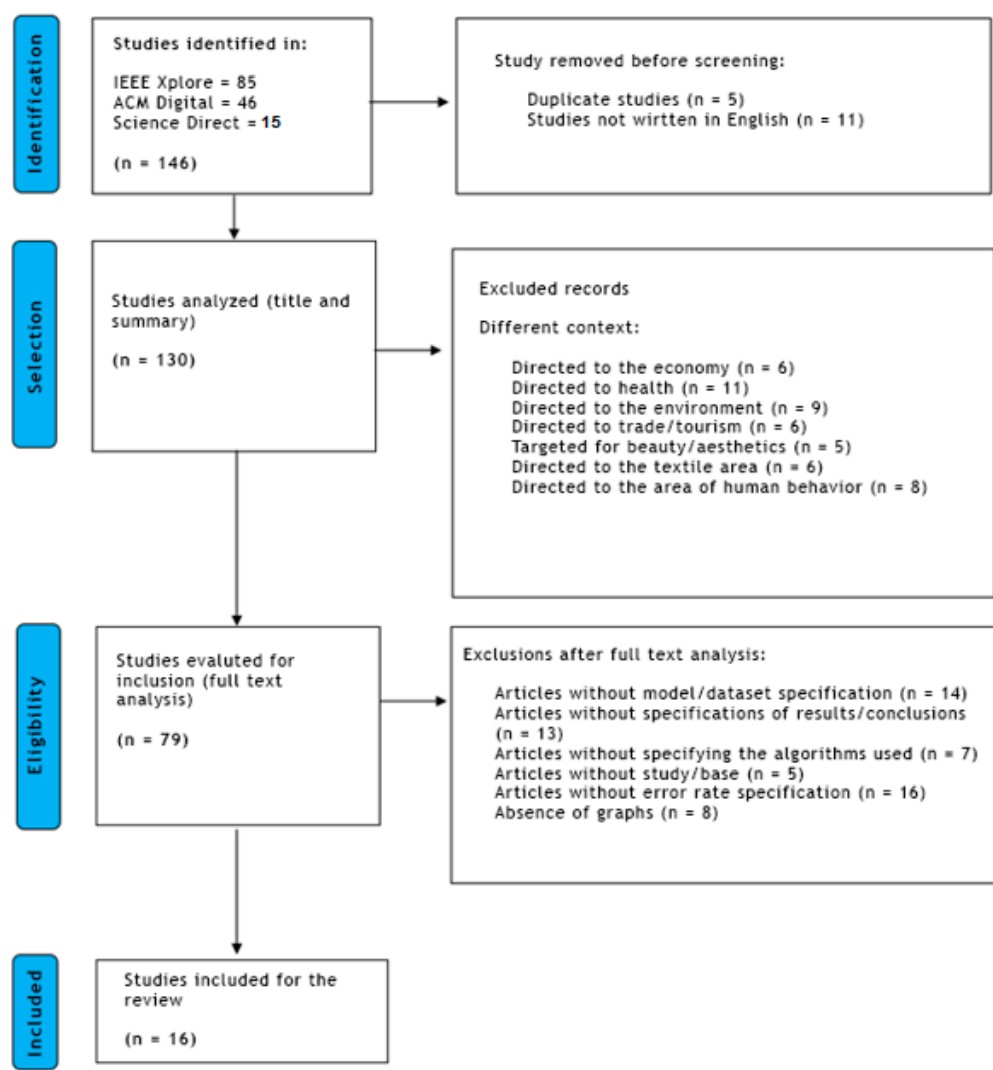

**Figure 2.** Flowchart of research phases.

Data were extracted from all studies identified using a predefined format. Data extracted included: study; model; description. Table 1 identifies the extracted data from the included studies.

**Table 1.** Scientific articles analyzed.

| Study | Model | Description |
|---|---|---|
| [8] | Inception-v3 Inception-v4 | The authors developed a system that exploits the extensive use of mobile devices to provide health information about foods consumed. The application is based on the recognition of a meal, and it presents the ingredients and their nutritional values. |
| [9] | ResNet-50 Inception V3 SENet-154 | The authors developed a system for recognizing ingredients and recipes, focusing on the analysis of some issues such as the recognition at image level of certain ingredients of a recipe, that is, the detection of the ingredients composed by the recognized recipe, and grouping in single or multiple image scaling. |
| [10] | AlexNet CaffeNet ResNet | The authors developed a framework, which they called DeepFood, that extracts effective features from a dataset of ingredient images using Machine Learning, which improves the average accuracy of multiclass classification by applying several advanced machine learning techniques. |

**Table 1.** *Cont.*

| Study | Model | Description |
|-------|-------|-------------|
| [11] | MatConvNet | The authors propose the use of neural networks to jointly consider food recognition, recognition of ingredients, and recognition of preparation methods, and verify that the recognition performance can be improved by considering several factors. |
| [12] | ResNet 101 NeuMF | The authors propose a solution for the recognition of multiple ingredients in the supermarket, using the Spatial Regularization Network (SRN) model to solve it. The authors also present a method for personalized recipe recommendation combined with the recognition of multiple ingredients. |
| [13] | ResNet-50 | The authors propose a system for the recognition of a recipe and, from the corresponding image, present the process of cooking the recipe, including the whole process and instructions for preparation and ingredients. |
| [14] | ResNet-50 | The authors analyzed a cross-modal recognition task between cooking recipes and food images and proposed an Adversarial Cross-Modal Embedding (ACME) framework to solve the recognition task in food domains. The main objective was to create an end-to-end ACME framework between cooking recipes and food images. |
| [15] | VGG ResNet DenseNet | The authors propose a Multi-Scale Multi-View Feature Aggregation (MSMVFA) scheme for food recognition that aims to detect and capture ingredients in each recipe. The MSMVFA method utilizes supervised convolutional neural networks (CNNs) to identify each type of ingredient and subsequently fuse them to obtain a more robust and discriminative representation of the recognized recipe. To validate the effectiveness and robustness of the proposed method, the authors implemented three models: VGG, ResNet, and DenseNet. |
| [16] | RESNet-50 SE-ResNet-50 CBAM-ResNet-50 | The authors of the ninth article refer to the difficulties of recognizing recipes or foods through applications. Therefore, in the article, its authors propose a Multi-View Attention Network (MVANet) within a multi-task framework that incorporates several semantic features in the task of food recognition, both in ingredient recognition and in the creation of recipes. The authors also propose a novel Multi-View Attention Fusion (MVAF) that serves to extract and fuse multiple semantic features from different tasks. In terms of implementation, the authors used deep neural networks in the MXNet software, where they implemented all the models for testing, including ResNet-50, E-ResNet-50, CBAM-ResNet-50, and the authors' model MVANET50. |
| [17] | VGG Arch-D AlexNet | The authors propose architectures for simultaneously recognizing ingredients and performing ingredient recognition and food categorization, exploring the mutual relationship between them. In terms of architecture, the authors performed a modification to the DCNN model, using four different Deep Learning architectures named Arch-A, Arch-B, Arch-C, and Arch-D. These architectures were modified from the 16-layer VGG model. |
| [18] | VGG | The authors developed a mobile application called Pic2Dish, which automatically recognizes the name of the dish and recommends a customized recipe along with instructional videos on how to cook the recipe. For the recognition module, the authors used a convolutional network through the VGG model. |
| [19] | RESTFul | The authors propose a system design and implementation that employs Deep Learning to simplify the cooking task. The system architecture relies on four modules: user-facing interfaces, an image recognition model, a search and recommendation engine, and a dataset. RESTful services are used to implement this architecture. |
| [20] | RESNet-50 | The authors propose a method that estimates the nutritional information from the detection of food ingredients in a given recipe. The authors present a model that estimates the nutritional value through semantic segmentation, extracting a label histogram, which expresses the frequency of occurrence of ingredient text tags based on the results of an image, and then estimating the nutritional value using regression analysis based on the previous information. |
| [21] | Cloud Broker | The authors propose a cloud-based mobile eHealth calorie system that classifies an image of food and further calculates the calories of each food with high accuracy. In this paper, the novelty is that the authors apply an intelligent cloud broker mechanism to efficiently use an instance that is managed in the cloud, which will help for better performance and a better and more efficient response time. |

**Table 1.** *Cont.*

| Study | Model | Description |
|---|---|---|
| [22] | AlexNet | The authors of the article used a convolutional neural network (CNN) for detecting and recognizing food images. They optimized the hyperparameters of the CNN and demonstrated that it significantly improved the accuracy of food recognition compared with a conventional method using a support vector machine (SVM). Through observation of the CNN they produced, the authors found that color features play a dominant role in the food recognition process. |
| [23] | Inception-v3 | The authors present a customized healthy food recommendation scheme, which personalized healthy food recommendation scheme, which they called Market2Dish. An Inception-v3 Net was used to perform the food image capture, reaching an accuracy of 95.5%. In terms of health profiling, the authors propose a supervised neuronal model to trace the health profile of each user by means of the content generated by that same user on social networks. |

According to the analysis of the selected articles, it is evident that there are numerous tools and solutions available for recognizing specific foods or recipes. One of the most crucial aspects of the studies reviewed is the development or examination of models designed to identify or recommend healthier foods and recipes, as healthy eating plays an increasingly important role in people's well-being.

Regarding the timeframe of the studies analyzed, they were conducted between 2014 and 2021, with the majority being published in 2017 and 2019, as illustrated in Figure 3.

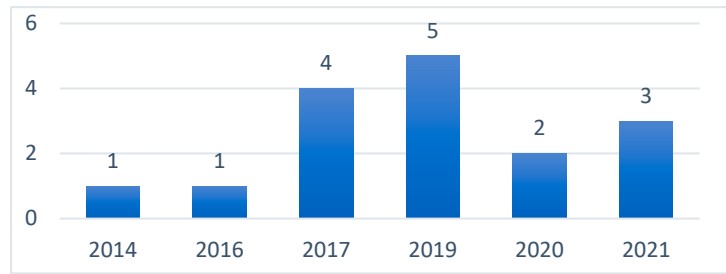

**Figure 3.** Date of publication of the articles.

The authors of these studies preferred to implement similar architectures, with 94% of the articles utilizing convolutional neural networks (CNNs) and only 6% using other deep learning methods, as shown in Figure 4. CNNs are deep learning models designed for automated end-to-end prediction. Therefore, it can be concluded that all architectures used in the studies analyzed employed CNN models, with different networks used for specific purposes in each article.

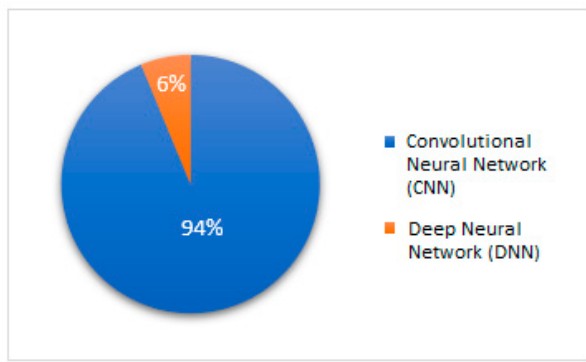

**Figure 4.** % of the architectures used in the articles.

As illustrated in Figure 5, the model used in the implementation of each study was quite diverse, since some authors chose to use only one while others preferred to implement several models to show which one was the best.

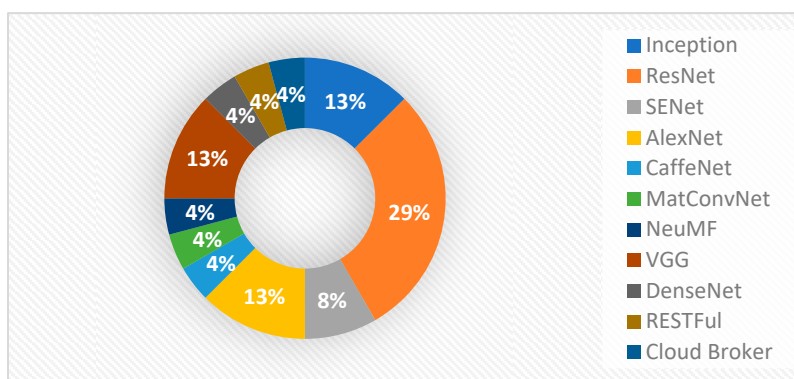

**Figure 5.** % of the architectures used in the articles.

In terms of the dataset used in the studies analyzed for analyzing the models and tests of their functioning, the authors used different alternatives, as can be seen in Figure 6. Some authors used more than one dataset to test the models implemented and verify and compare the results obtained. Some authors utilized multiple datasets to test their implemented models and compare the results. The most commonly used dataset was VIREOFood, which many authors considered comprehensive and complete in terms of food categories.

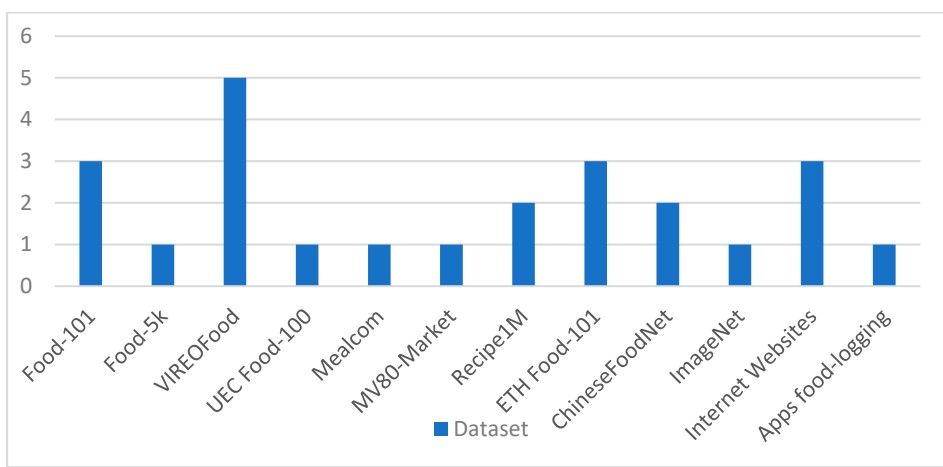

**Figure 6.** Quantity of datasets used in the articles.

It is worth noting that three (3) of the articles used websites to collect information and images of food or recipes from users, while one (1) article used a food-logging app to create a dataset using high-quality smartphone images. In conclusion, this study revealed that various models can be utilized for food recognition and recommendation, which can be implemented using convolutional neural networks or Deep Learning. However, only three of the articles provided information regarding the software used for implementation, namely Keras, PyTorch, and MXNet.

Although there are many articles on the subject, most of them are very similar in their studies and developments. There is a lot of similarity in the studies and developments; however, there are many articles that contain few results or do not mention the method of development used. This is proven by the analysis of their contents: of one hundred and forty-six (146) articles, only seventeen (16) showed to have results, important content, and to be contained within the scope of the intended theme.

After the analysis of the scientific articles and verification of their results, it is concluded that, besides the implementation method being one of the most important, it is fundamental to make a good choice of the software to be used, the architecture, and the most accurate models for recognizing a given food and recommending recipes. It is also possible to see that different types of models and architectures can be used to carry out the recognition and recommendation of recipes, and this implementation requires programming that is too complex to be solved using traditional programming since it is sometimes necessary to make changes in the architectures or models.

Based on the above information, the answer to the research questions is presented:

(a)  Question 1: Are there digital solutions that offer recipe proposals based on ingredient recognition or ingredient images?

As the proposed topic is comprehensive and image recognition is becoming increasingly common in the food industry, many people use mobile applications to simplify tasks. Among the scientific articles studied, various solutions for ingredient recognition using different implementation methods were found, as referenced in articles [8–23]. Although no article specifically mentions the recognition of food packages, some articles tested image datasets that included images of packaged food, such as the VireoFood dataset. This dataset contains pictures of recipes, foods, ingredients, and some packaged foods; it is mentioned in articles [9,15–18].

(b)  Question 2: What are the frameworks for ingredient recognition solutions?

Frameworks are interfaces that contain a set of generic codes, enabling the development and implementation of models in a fast and effective way, thus reducing the time required to reproduce a certain function. Concerning the intended topic of food or ingredient recognition, several available frameworks can perform this function, such as PyTorch, as referenced in the article [13], MXNet in the article [16], and Keras in the article [19].

(c)  Question 3: Are there digital solutions for recipe proposals based on leftover meals to fight food waste?

Food waste is a growing concern in today's world due to high consumption and variety, which has serious social, economic, and environmental consequences. Despite analyzing several articles, none of them directly address the problem of food waste. However, combating food waste can involve actions such as buying food that can be consumed within its shelf life, and recommending recipes based on the recognition of purchased foods is one way to do so. The articles that focus on recipe recommendations based on food or ingredient recognition are [12,18,19,23].

After a comprehensive study, it was concluded that the ResNet-50 model is the most frequently used and mentioned in the scientific articles analyzed. Additionally, the Food-101 dataset is the most cited and utilized for tests and results, despite being extensive and primarily consisting of images with several ingredients and fewer images of simple ingredients.

## 3. Materials and Methods

The work has been divided into two parts. The first part aims at food ingredient recognition through a given dataset. The second part is the recommendation of recipes based on the food ingredients recognized in the first part.

### 3.1. Proposed Model for Ingredient Identification

For the first part of food ingredient identification, a convolutional neural network (CNN) model was created. In this case, the pre-trained model ResNet-50 was used due to its frequent use in scientific articles and being one of the most commonly used models when the objective is the classification or recognition of a given image. The ResNet-50 model is a variant of the original ResNet model. Although both models use the same processing concept, the difference lies in the number of layers; while the ResNet-50 uses 50 layers in its network, the ResNet model uses 34 layers. The architecture of the ResNet-

50 model consists of a sequence of convolutional blocks, which are 5 blocks [24]. The ResNet-50 model consists of 48 convolutional layers; there is also one MaxPool layer and one AveragePool layer.

Initially, the project takes an input that can have a size of either $224 \times 224$ or $128 \times 128$. For this implementation, $128 \times 128$ inputs are used because they require less memory and processing time than $224 \times 224$ inputs. The entire ResNet architecture includes the initial layers, such as the convolution and max-pooling layers. Next, the neural network begins with the first state, state 1, which contains three blocks of layers. The identity block [25] is a standard block used in the ResNet neural network. It corresponds to the case where the input value is equal to the size of the output value. If the size is different, a convolution block is used to make the input value equal to the output value. The ResNet model uses the concept of skip connection [26], which is to add the original input value with the output value of the convolution block. This skip connection is a very important concept in the ResNet model [25].

As each state transition occurs, the window size doubles while the input size halves. Finally, the network includes an Average Pooling layer, followed by a Fully Connected layer with a softmax activation function and 1000 neurons for classification. The ResNet-50 model has about 23 million trainable parameters. Additionally, two convolutional neural network layers were added to enhance the model's accuracy level, which will be further elaborated during the testing phase using the chosen dataset.

The batch size used was 32, which means that during each training iteration, the model will process and update its parameters using 32 training examples at a time. This batch size was chosen because batch size 32 is generally the most efficient in terms of computational usage, meaning it does not require much memory and processing. In terms of the optimizer, "adam" optimizer was selected because it is one of the most widely used and because it adjusts the learning rates individually for each parameter of the model, adapting them according to the average of the gradients. Regarding the learning rate, since "adam" optimizer has been used, this optimizer has a learning rate set to a value of 0.001. This learning rate provides a good balance between fast convergence and stability.

### 3.2. Dataset

Datasets are specific databases that serve as samples for training artificial intelligence algorithms or other types of Data Science projects. The selection of the right dataset is crucial for training a model that produces accurate results. After analyzing scientific studies on similar themes, it was highlighted that the Food-101 dataset is a popular choice. Despite this, the dataset is extensive, with around 101,000 images that mostly contain images of prepared recipes, which do not align with the project's main objective of using simple ingredients. Therefore, a search for alternative datasets was conducted, and the "Fruits and Vegetables Image Recognition Dataset," available on the Kaggle website, was found to be more relevant to the project's objectives. However, it restricts the range of ingredients to fruits and vegetables only.

The chosen dataset contains three different folders with similar images, as seen in Figure 7. This feature allows for different ways of implementing the model, which will be explained later in detail with the help of images of the implemented code. It is worth noting that datasets are generally not divided, as this division operation is performed through a code block. However, the dataset used in this project is already divided, so there is no need to perform this division in the code.

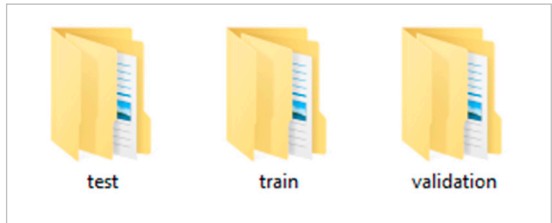

**Figure 7.** Dataset folders.

The train folder contains all the images in the dataset, i.e., the images that the model will use to train and learn during the process. The validation folder is used to evaluate the model; this way, it provides an unbiased evaluation of the model on the training dataset while adjusting the model hyperparameter. The results obtained through the images in the validation folder are used, and from those, the hyperparameter is increased, thus indirectly affecting the model. Its main objective is to verify that the model is not overfitting (Overfitting is described when a model fits a given set of previously observed data very well but is not effective in predicting new outcomes).

The test folder's function is to evaluate the final model based on the set of images in the train folder, determining the level of accuracy of the final model. Within each of the above-mentioned folders, there are the same number of classes, 36 classes of food ingredients, and within each of these classes are the images of the corresponding ingredients, illustrated in Figure 8.

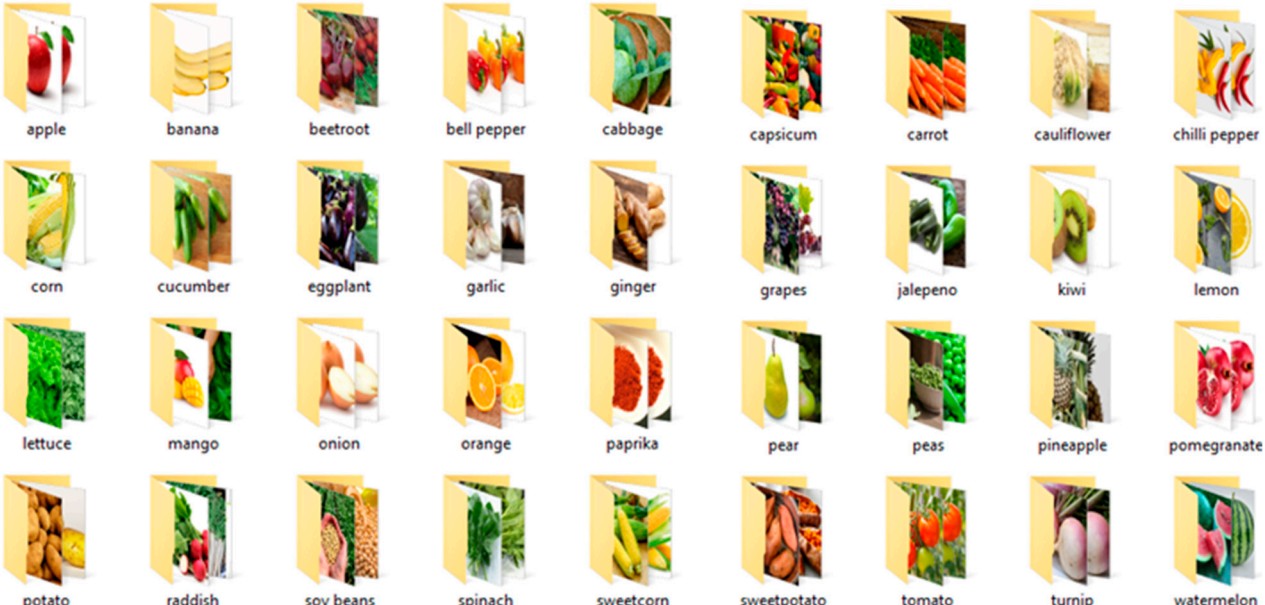

**Figure 8.** Folders of the dataset classes.

The resolution of the images in the "Fruits and Vegetables Image Recognition Dataset" available in Kaggle is diverse, as it depends on the specific version of the dataset and the sources of the images. However, the dataset presented contains a set of images with different resolutions.

Example: In this example, one of the images from the dataset is verified as present in the train folder and has a size of 2000 × 1000, as shown in Figure 9.

```python
from PIL import Image

# Path to the directory containing the dataset images
dataset_dir = "C:/Users/rodri/Desktop/Dataset/train/banana"

# Load an example image from the dataset
image_path = dataset_dir + "/Image_27.jpg"
image = Image.open(image_path)

# Get the resolution of the image
image_width, image_height = image.size
print("Image resolution:", image_width, "x", image_height)
```
```
Image resolution: 2000 x 1000
```

**Figure 9.** Image resolution example.

So, the characterization of the dataset used:

- 36 classes of food ingredients;
- 3115 images in total;
- Fruit and vegetable images;
- Images in PNG image format;
- Different sizes and resolutions;
- Images with different angles, types of luminosity, and quality.

### 3.3. Results and Discussion

Model construction of this project was based on the ResNet-50 model, created from the tool available in Anaconda software [27], the Notebook Jupyter [28], which, through a set of iterations, was successively adjusted, having chosen the best layers and activation functions to obtain better accuracy. The specifications of the computer, software, and libraries are: Computer: SO Windows 10 Home, version 22H2, Intel Core i7.4510U, 8 GB RAM, 64-bit operating system; Software tools used: Anancoda Navigator 2.4.1, Jupyter Notebook 6.5.4, Spyder 5.3.3; Libraries used: Matplotlib 3.4.3, Numpy 1.21.5, Pandas 1.3.4. Previously, as mentioned, a dataset "Fruits and Vegetables," which contains a total of 3115 ingredient images, was selected, which will aim to train the CNN model with the dataset images and predict the category of the recognized ingredient. Firstly, after creating a new file in Notebook Jupyter, the first blocks of code are related to the inclusion of the necessary libraries for the implementation of the model, like Keras, Tensorflow, Pandas, Numpy, and Matplotlib.

Next, data augmentation (Data Augmentation is a set of techniques for artificially augmenting a quantity of data by adding slightly modified copies from existing data, thus increasing the diversity of the data in a given dataset [29]) was implemented, which aims to perform rotation and orientation modifications of the dataset images, allowing the CNN model to perform the learning process of the same image in different positions, as illustrated in Figure 10.

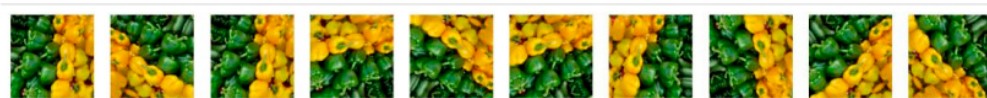

**Figure 10.** Data augmentation in an image.

The ResNet-50 convolutional neural network model was modified in the final phase by adding four layers: a Flatten layer, an Activation layer with the ReLu function, a Dense layer, and an Activation layer with the Softmax function. These additional layers were added to improve the accuracy of the CNN model.

Once the model was created with these modifications, training was conducted to verify its accuracy, with only 4 epochs (Epochs is a hyperparameter that defines the number of times the learning algorithm will traverse through the entire training data set) defined for the initial test. It is important to note that this number of epochs is relatively low, as the

objective of this phase is to test the basic functionality of the model. The number of epochs required can vary depending on the dataset, with larger datasets requiring more epochs.

The accuracy value, measured in percentages, represents the percentage of images correctly classified by the model after training. The error value indicates the level of accuracy of the model in classifying images.

The Matplotlib library was used to visualize the precision of hit-and-miss over epochs. Figure 11a shows the accuracy of the model, with the hit value increasing as the number of epochs increases, achieving approximately 92% accuracy on the training dataset (accuracy) and 83% on the test dataset (val_accuracy). Figure 11b displays the behavior of the model in classifying images, with the value of losses decreasing over the epochs. It is worth noting that the training losses (loss) are lower than the validation losses (val_loss). Also, in the Figure 11c is displayed the accuracy using 4 epochs, and in the Figure 11d is displayed the loss using 4 epochs.

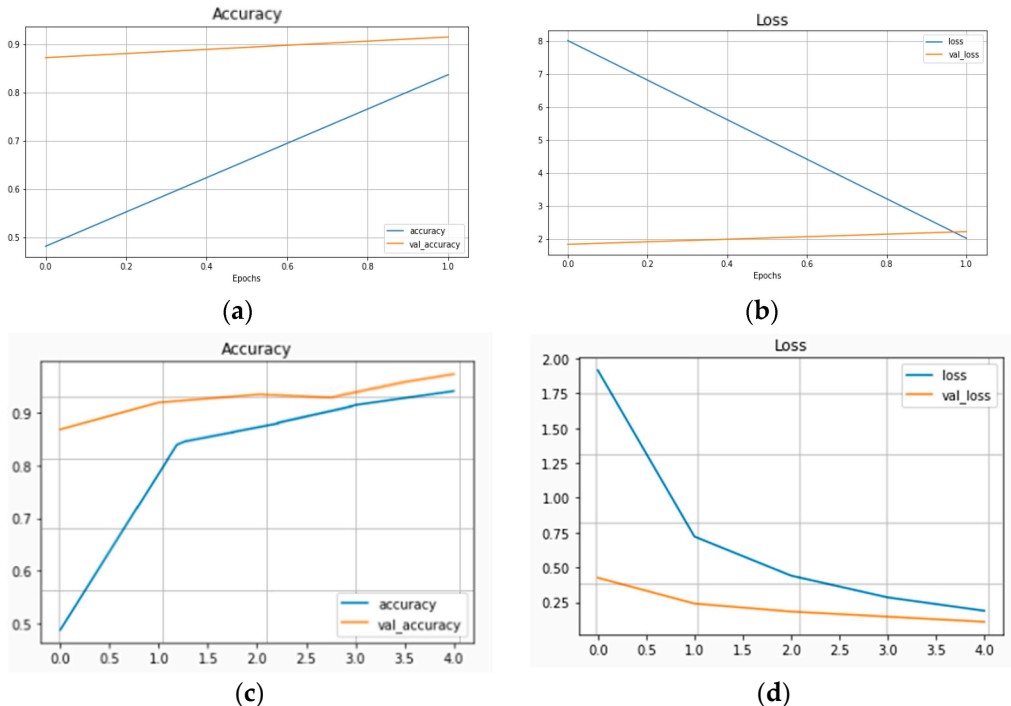

**Figure 11.** (**a**) Graph of the accuracy of the first test; (**b**) Graph of the losses of the first test; (**c**) Graph of the accuracy of the test with 4 epochs; (**d**) Graph of the losses of the test with 4 epochs.

A final value for the model's hit level was generated by testing the model with multiple epochs and previous values. The overall hit value reached approximately 90% after two epochs, indicating that the model has a very satisfactory hit rate for this initial phase. Additionally, the model has a satisfactory accuracy rate during this phase.

To conclude, a test was performed within the Jupyter Notebook tool to verify the effectiveness and accuracy of the model on specific images. A small piece of code was created to select ten images and compare the model's prediction with the actual class of each image. Captions were added to each image to show the prediction and its class. As shown in Figure 12, the model correctly classified all the images, demonstrating its high accuracy in recognizing the corresponding classes.

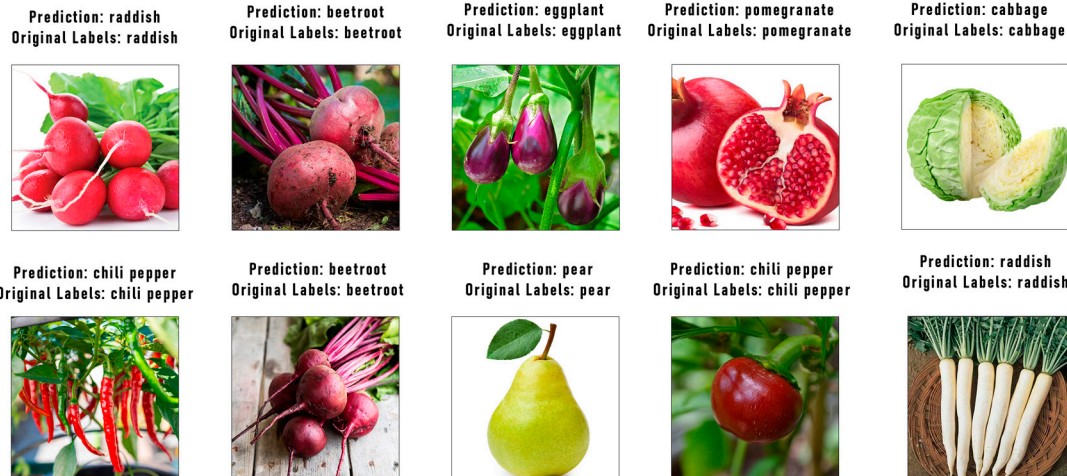

**Figure 12.** Verification of model accuracy.

Subsequently, only one change was made in the model implementation to improve the results. The change was made by increasing the number of epochs to 12 for the model training to verify if the model would achieve more accurate image classification compared with the values obtained in the first test.

It should be noted that the number of epochs could not be much larger due to the limited size of the dataset, which contains only 3115 images. Using a very high number of epochs can lead to overfitting of the model, resulting in repeated and incorrect final values and thus negatively impacting the model's performance.

In this final test, the value of correct answers increased gradually, remaining constant from the seventh epoch on, obtaining a hit rate of about 96%. As for the validation values (val_accuracy), they were gradually increasing, reaching a value of about 95%, as illustrated by Figure 13a. The graph in Figure 13b confirms that the loss values have been decreasing, both in the test values (loss) and in the validation values (val_loss). In the last epoch, there was a slight increase in the test values and a significant decrease in the test values. epoch. The value obtained for training loss increases after epoch 7. One of the reasons already mentioned in Section 3.2 is overfitting, which means that the model becomes too specialized in capturing the patterns in the training data and cannot generalize under certain conditions, especially when the dataset is small. With a smaller dataset, the number of epochs needed for the model to learn the patterns in the training data is also smaller. In other words, by increasing the number of epochs, the model will eventually overfit. In this case, what may be happening is exactly this: at epoch 7, the model has already recognized all the patterns. In this case, it does not represent a significant problem and can be solved with a model layer adjustment or using a larger dataset.

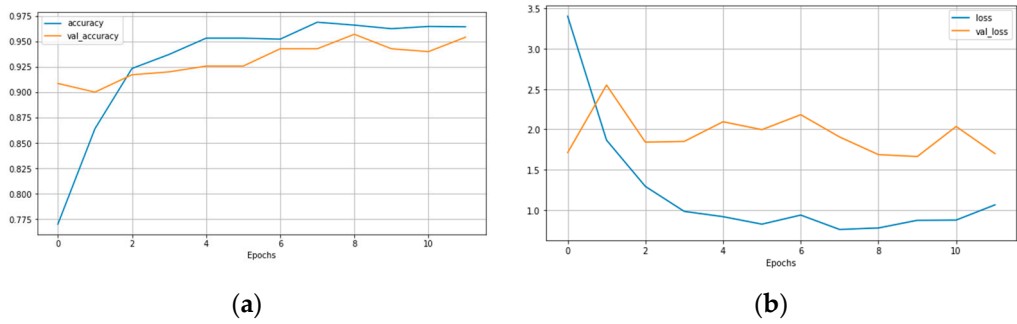

(a)          (b)

**Figure 13.** (**a**) Accuracy plot of the second test; (**b**) Loss graph of the second test.

Through this final test, it was confirmed that the choice of the ResNet-50 model is a suitable choice for the realization of the project, allowing it to fulfill the main objective of ingredient recognition, obtaining approximately a value of 96% accuracy and a loss rate of about 4%. As the results obtained were quite positive, we proceeded to the final step, saving the model we had created and trained.

In the confusion matrix, illustrated in Figure 14, we can observe that the decision to exclude certain classes is driven by the need to prioritize the evaluation of specific classes that hold greater significance or relevance. By selectively choosing classes for analysis, we can concentrate our attention on those that are most critical. For example, we can verify that the model had difficulty predicting the bell pepper class since many times it predicted that the image was capsicum. The model is not making a bad prediction since bell pepper and capsicum are the same ingredients; they just have different names in certain parts of the world. In this case, it is a good example of how the confusion matrix is a way to check the accuracy and sensitivity of the model.

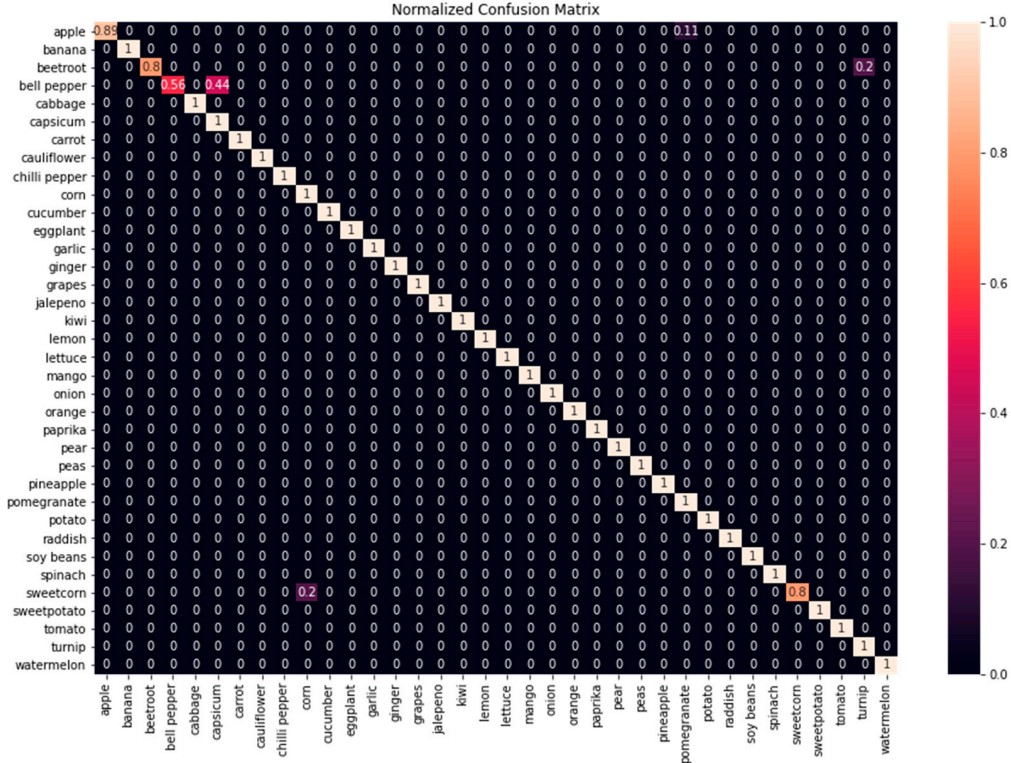

**Figure 14.** Confusion matrix.

## 4. Deploying the Model

The described part in Section 4 is aimed at recognizing the ingredients, which involves the creation of a convolutional neural network, using the ResNet-50 model, which was previously trained with the described dataset. Then proceed with the main objective, a recipe recommendation that contains the recognized ingredient.

### 4.1. Workflow

In this way, we proceeded to create a web solution that served as a proof of concept, allowing the user to select the image of the ingredient, and then this image would be analyzed by the recognition model previously developed, and then the visualizations of recipes in which the recognized ingredient is used would be able to visualize its complete ingredients and instructions. The architecture of this solution is schematized in Figure 15.

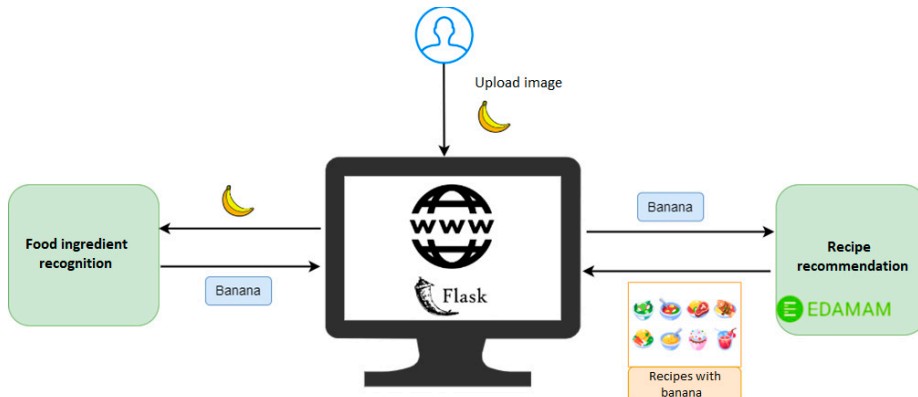

**Figure 15.** Project workflow.

To create this solution, we used the Flask framework through the Spyder software, and an API called Edamam [30] was used to recommend recipes.

### 4.2. Image Recognition

The goal of this food ingredient recognition component is to recognize the ingredient desired by the user so that it is possible to identify which ingredient it is and, later, which recipes contain that recognized ingredient. There are many ways of recognizing ingredients, such as video recognition, voice input recognition of the ingredient, and recognition of images of ingredients. In this paper, we explore the possibility of ingredient recognition using images or photos given by the user.

In Figure 15, the image input is given by the user through an upload option on the homepage of the web application. Next, the ingredient will be recognized through the implemented model, ResNet-50.

To perform the task of recognizing food ingredients, the Flask micro-framework was utilized. Flask is a framework that allows for the creation of web applications and is widely used for implementing CNN models. Therefore, a web application was developed using the Flask framework to implement our CNN model, with two routes established:

- Route 1: An index page that allows users to upload an image of the food ingredient, as shown in Figure 16.
- Route 2: A prediction page that recognizes the food ingredient image and sends it to the created model to determine the food ingredient class. This information is then returned to the webpage, as shown in Figure 17.

### 4.3. Recipe Recommendation System

To make the recommendation of recipes based on the ingredient chosen by the user, the JavaScript language was used to search the database, API Edamam, by the recognized ingredient to return the recipes that contain the recognized ingredient.

We have also made some options for the user, in which the user can choose diets or preferences from the recommended recipes, as shown in Figure 17. From the options selected by the user, the program will perform a search in the database, returning the result of the recognized ingredient together with the diet options and preferences chosen by the user. The options available to the user on the form for diets are High protein; Low fat; Low carbohydrate; High fiber content; and Balanced. And for available preferences: None; Reduced sugar; No alcohol; Peanut-free; Nut-free; and Vegetarian.

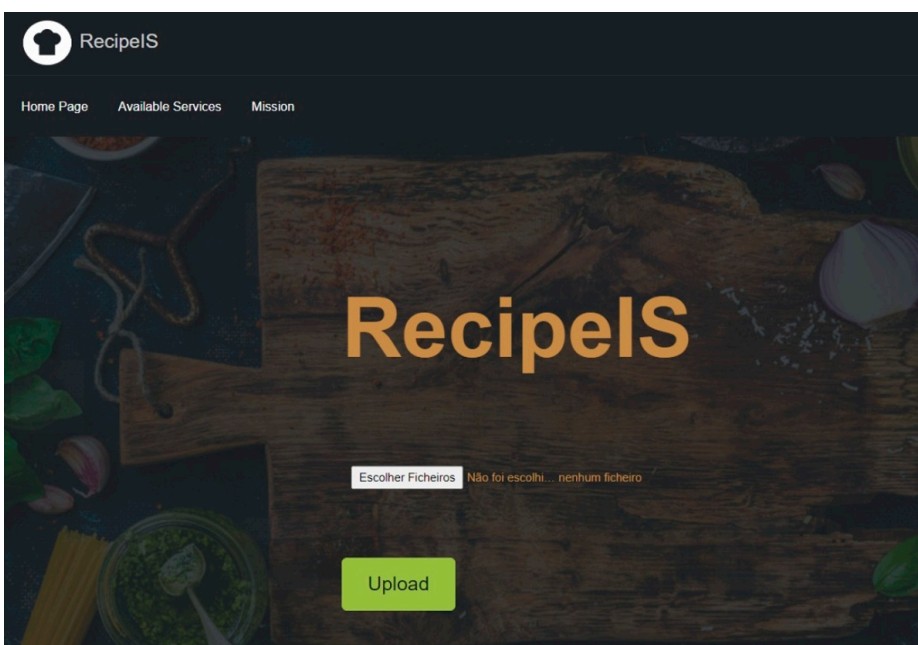

**Figure 16.** Print of the index page with image upload option for the user.

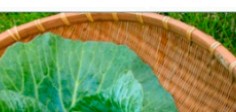

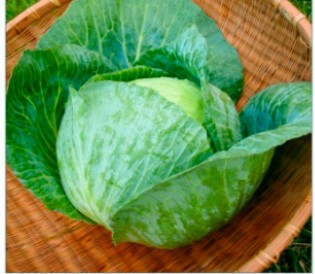

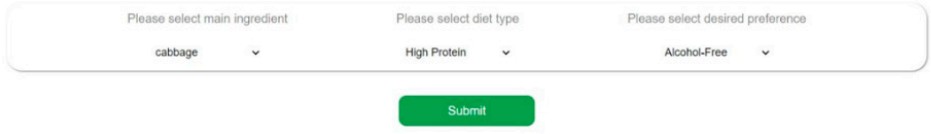

**Figure 17.** Print of the recognized ingredient with diets and preferences.

Finally, the recommended recipes are returned, which contain all the ingredients needed to make the recipe and the link that allows the user to access the website of the selected recipe to obtain the instructions and more information about the particular recipe, as illustrated in Figure 18.

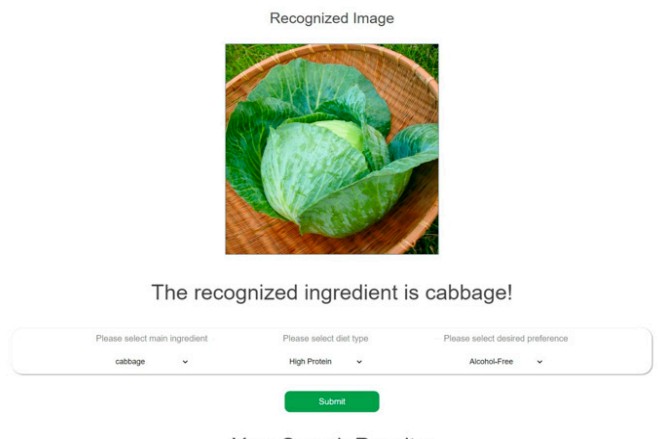

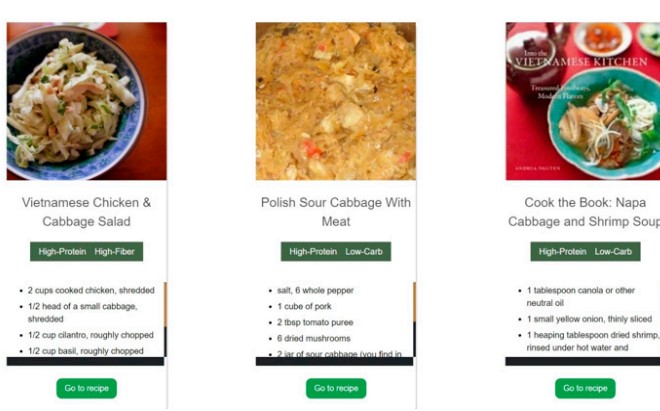

**Figure 18.** Print of the predicted webpage with the recipe recommendations (in this example, recipes with cabbage as the main ingredient).

## 5. Conclusions

This paper presents a convolutional neural network model for ingredient recognition and, as a proof of concept, the recommendation of recipes based on the recognized ingredients. The proposed model is ResNet-50, to which layers were added in the final part, and a dataset with 36 food ingredient classes was used. The model was trained with this dataset and obtained different results in two tests, as mentioned in Section 3.2. The first test achieved an accuracy rate of 90%, while the second achieved 96%, proving the model's adequacy for the project's development.

In the recommendation task, the use of the Edamam API was an important choice as it enabled recipe recommendation without the need to create a database. It was also equally effective and allowed for the inclusion of filters, providing users with a more personalized selection.

For future work, it would be important to use a larger training dataset with more food images and classes to access a larger set of recipes. Another feature for a more commercial solution would allow users to upload multiple ingredient images, enabling the program to recognize several ingredients simultaneously and recommend recipes with the combined ingredients. It would also be interesting to add or remove ingredients to recommend more specific recipes. Creating a mobile application with the same purpose would be beneficial, allowing users to receive real-time ingredient image recognition and recipe recommendations.

It is worth noting that the project employed the Edamam API as a database instead of developing a custom database from scratch. This decision not only eliminated the need for extensive programming but also introduced a novel approach to recipe recommendations. Through thorough research and investigation, it was discovered that harnessing

the capabilities of the Edamam API proved to be a promising and efficient method for generating recipe recommendations. That said, as future work, it is recommended to explore the integration of recommendation systems into the project to enhance the user experience and make it more complete overall. Some examples of recommendation systems that could be searched for are collaborative filtering-based systems that leverage the preferences of similar users to provide insightful recommendations. By making these investigations and implementing them, the objective would be to make a more effective recommendation system.

**Author Contributions:** Conceptualization, Methodology, M.S.R., F.F. and Â.O.; Software, Validation, Investigation, Resources, Data curation, Writing—original draft, M.S.R.; Writing—review and editing, M.S.R., F.F. and Â.O.; Supervision, F.F. and Â.O. All authors have read and agreed to the published version of the manuscript.

**Funding:** This research received no external funding.

**Institutional Review Board Statement:** Not applicable.

**Informed Consent Statement:** Not applicable.

**Data Availability Statement:** Not applicable.

**Conflicts of Interest:** The authors declare no conflict of interest.

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
