# Peer review of "RecipeIS—Recipe Recommendation System Based on Recognition of Food Ingredients"

_applsci, doi:10.3390/app13137880_

Round 1
Reviewer 1 Report
The authors have addressed an interesting problem of recipe recommendation based on the food items available to an individual. However, most of the paper is dedicated to fruit and vegetable identification problem with few details of the recommendation system. If all recipes are stored in a database, and the ingredients are used as input queries to fetch the related records of the recipes from this database, then it is not a recommender system per se. The authors need to develop a recommender system using one of the various approaches available, such as collaborative, content-based, hybrid, etc. The design details and choices must be clearly defined, the proposed technique must be evaluated using any recommender system evaluation techniques, and results must be compared with the state-of-the-art. It is also pertinent to mention that food recipes may need other items besides fruits and vegetables, such as spices, accessories, and dairy items (such as eggs, cheese, butter, etc.). A comprehensive system must consider recipes wholistically for a useful application. I leave the final decision to the authors, but making a pipeline of ingredient identification and recipe recommendation seems overkill. A technique may be developed wherein a user is required to give the list of available items (using text or from given images), and the system may generate recipes based on this list of items to make it an effective recommender system.
Although some minor editing might be required, the overall quality of the English language is acceptable.
Author Response
The authors have addressed an interesting problem of recipe recommendation based on the food items available to an individual. However, most of the paper is dedicated to fruit and vegetable identification problem with few details of the recommendation system. If all recipes are stored in a database, and the ingredients are used as input queries to fetch the related records of the recipes from this database, then it is not a recommender system per se. The authors need to develop a recommender system using one of the various approaches available, such as collaborative, content-based, hybrid, etc. The design details and choices must be clearly defined, the proposed technique must be evaluated using any recommender system evaluation techniques, and results must be compared with the state-of-the-art. It is also pertinent to mention that food recipes may need other items besides fruits and vegetables, such as spices, accessories, and dairy items (such as eggs, cheese, butter, etc.). A comprehensive system must consider recipes wholistically for a useful application. I leave the final decision to the authors, but making a pipeline of ingredient identification and recipe recommendation seems overkill. A technique may be developed wherein a user is required to give the list of available items (using text or from given images), and the system may generate recipes based on this list of items to make it an effective recommender system.
- We would like to thank you for the insightful comments and recommendations.
We agree with the comments however, at this stage, the idea was to focus the work on recognizing an ingredient, generating recipes according to the user's preference, with the recipe based on the recognized ingredient. As a proof of concept, a dataset containing just fruits and vegetables was used.
Considering your indication, the next step for this work would be the systematic review of recommendation systems, identifying the most suitable for this reality. This aspect was included in the "Conclusions" section, where the recognition of a set of ingredients is also identified as future work.
Although some minor editing might be required, the overall quality of the English language is acceptable.
- We will send the manuscript for proofreading by a native speaker.

Reviewer 2 Report
General Comments:
1) The content of the manuscript conforms to the scope of Innovative Technology in Food Analysis and Processing.
2) Authors have proposed an recipe recommendation system based on recognition of food ingredients. The ingredient recognition system based on images and ResNet50 has attained accuracy of 96%.
3) The write-up of the Introduction are not focused. Authors should discuss and address the gap in the literature. Then, dwell on the issue of ingredient leftovers, and recommend recipes to avoid wastage.
4) Issues with Netflix and Youtube recommendations are not relevant and should be removed. The limiting factor to the recipe recommendation in this context is the leftover ingredients.
5) While the concept seems feasible, the authors seemed to have missed out on several research design aspects.
6) The aim of the the recommendation system is to generally avoid wastage of resources. Hence, the system is flawed as it does not take into consideration on quantity left for each ingredient.
7) The different composition of the leftover ingredients will affect the recipe being recommended. However, this critical control element is missing.
8) Therefore, the system is not really different than others accept that it recognizes the ingredients from images, and uses that to provide a general recommendation of recipes.
9) The general recommendation of recipes defeats the purpose of making use of leftover ingredients, and does not specifically answer the issue of wastage.
10) A more impactful approach to this is to develop a model that can recognize the amount of ingredient leftovers. By knowing the proportion of each ingredient, the model can suggest more practical recipe.
11) The systematic review pointed out ResNet as good CNN architecture for recognition of food ingredients. Apart from being frequently used, what makes it more desirable than simpler structures like AlexNet?
12) A total of 27 references have been cited throughout the manuscript. However, 10 of them are not from refereed journals or proceedings. Please replace those references with more reputable sources.
Specific Comments:
1) Based on the description on the inclusion criteria, Criteria 4 and 5 is applied earlier. Hence, it should be made Criteria 3 and 4. Meanwhile, Criteria 3 is applied later and should be labelled as Criteria 5.
2) Question 2 of the research questions are not relevant, as the focus of the system is to recommend recipes based on ingredient leftovers. The question should be removed as it is not the main focus of the study.
3) In 3.1 Proposed Model for Ingredient Identification, please rectify (line 228) "convulsion" to "convolution".
4) Please include the hyperparameter settings such as batch size, optimizer and learning rate in 3.1 Proposed Model for Ingredient Identification.
5) In the caption for Figure 7 (line 247) and Figure 9 (line 250), please rectify "Adaptaded" to "Adapted".
6) The first line of the second paragraph in Page 12 must be indented (line 285).
7) Authors have not mentioned the specification of the images. Please include the dimension and resolution of the image samples at the end of 3.2 Dataset.
8) Also, state the specifications of the machine, software and platforms where the experiments are being performed.
9) The initial test is defined by 4 epochs. However, the results shown in Figure 13 only indicate 1 epoch. Please show the performance over 4 epochs.
10) The caption for Figure 15(a) (line 363) should be rectified from "Precision" to "Accuracy".
11) In Figure 15(b), the training loss increases after Epoch 7. Would this be a technical concern? If this can be neglected, please elaborate and justify.
12) There should be another section in Section 3 that describes on the performance metrics. Apart from accuracy, include other measures such as precision and sensitivity.
13) Please show the detailed results on the accuracy of ingredient recognition. Precision and sensitivity can measures can directly be derived from the confusion matrix generated from the experiments.
1) The writing style conforms to technical writing norms.
2) However, there are spelling errors that require corrections.
3) Please refer to "Comments and Suggestions for Authors
".
4) Authors should send the manuscript for proof-reading by a native speaker.
Author Response
We would like to thank you for the insightful comments and recommendations.
The manuscript was updated considering each one of the general comments.
For specific comments:
1) Based on the description on the inclusion criteria, Criteria 4 and 5 is applied earlier. Hence, it should be made Criteria 3 and 4. Meanwhile, Criteria 3 is applied later and should be labelled as Criteria 5.
- The manuscript was updated considering the change of criteria order.
2) Question 2 of the research questions are not relevant, as the focus of the system is to recommend recipes based on ingredient leftovers. The question should be removed as it is not the main focus of the study.
- The manuscript was updated by removing question 2.
3) In 3.1 Proposed Model for Ingredient Identification, please rectify (line 228) "convulsion" to "convolution".
- The manuscript was updated.
4) Please include the hyperparameter settings such as batch size, optimizer and learning rate in 3.1 Proposed Model for Ingredient Identification.
- The manuscript was updated with the hyperparameter settings.
5) In the caption for Figure 7 (line 247) and Figure 9 (line 250), please rectify "Adaptaded" to "Adapted".
- The manuscript was updated.
6) The first line of the second paragraph in Page 12 must be indented (line 285).
- The manuscript was updated.
7) Authors have not mentioned the specification of the images. Please include the dimension and resolution of the image samples at the end of 3.2 Dataset.
- The manuscript was updated with the specification of the images.
8) Also, state the specifications of the machine, software and platforms where the experiments are being performed.
- The manuscript was updated with the required specifications.
9) The initial test is defined by 4 epochs. However, the results shown in Figure 13 only indicate 1 epoch. Please show the performance over 4 epochs.
- The manuscript was updated with figures to show the performance over 4 epochs.
10) The caption for Figure 15(a) (line 363) should be rectified from "Precision" to "Accuracy".
- The manuscript was updated.
11) In Figure 15(b), the training loss increases after Epoch 7. Would this be a technical concern? If this can be neglected, please elaborate and justify.
- The manuscript was updated with information to elucidate the issue.
12) There should be another section in Section 3 that describes on the performance metrics. Apart from accuracy, include other measures such as precision and sensitivity.
13) Please show the detailed results on the accuracy of ingredient recognition. Precision and sensitivity can measures can directly be derived from the confusion matrix generated from the experiments.
- The manuscript was updated to elucidate the issue using a confusion matrix.
Comments on the Quality of English Language
1) The writing style conforms to technical writing norms.
2) However, there are spelling errors that require corrections.
3) Please refer to "Comments and Suggestions for Authors
".
4) Authors should send the manuscript for proof-reading by a native speaker.
- We will send the manuscript for proofreading by a native speaker.

Reviewer 3 Report
More illustration of the methodology used is required.
The novelty must be clearly stated.
Language must be relooked to make it more simpler sentences.
Author Response
We would like to thank you for the insightful comments and recommendations.
More illustration of the methodology used is required.
- A new sentence and a schema were added in the "Introduction" section.
The novelty must be clearly stated.
- A new sentence was added in the "Introduction" section.
Language must be relooked to make it more simpler sentences.
- We will send the manuscript for proofreading by a native speaker.

Round 2
Reviewer 1 Report
The authors have made some improvements in the paper. However, the following questions are still unanswered:
1. Food recipes usually need other items besides fruits and vegetables, such as spices, accessories, and dairy items like eggs, cheese, and butter. The scope of the current work is limited to fruit and vegetable identification. Hence, it cannot recommend a recipe based on the items available to a user.
2. Making a pipeline of ingredient identification and recipe recommendation seems overkill to me. A user may give a list of available items and find food recipes through, let’s say, Edamam.
Although some minor editing might be required, the overall quality of the English language is acceptable.
Author Response
We would like to thank you again for the insightful comments and recommendations.
- Food recipes usually need other items besides fruits and vegetables, such as spices, accessories, and dairy items like eggs, cheese, and butter. The scope of the current work is limited to fruit and vegetable identification. Hence, it cannot recommend a recipe based on the items available to a user.
- Making a pipeline of ingredient identification and recipe recommendation seems overkill to me. A user may give a list of available items and find food recipes through, let’s say, Edamam.
R. The main objective of our research is to recommend a variety of recipes based on a recognized ingredient. Currently, we have a limitation of the recognition of only one ingredient at a time. However, as we mention in the "Future Work" section, our proposed improvement is to enhance the ingredient recognition system to handle multiple ingredients provided by the user.
We would like to point out that the use of the "Fruit and Vegetables" dataset, as discussed in section 3.2, where we add a sentence in the paragraph:
"Despite this, the dataset is extensive, with around 101,000 images that mostly contain images of prepared recipes, which do not align with the project's main objective of using simple ingredients. Therefore, a search for alternative datasets was conducted, and the "Fruits and Vegetables Image Recognition Dataset," available on the Kaggle website, was found to be more relevant to the project's objectives. However, restricts the range of ingredients to fruits and vegetables only."
To address this limitation, we believe it would be beneficial to test our model with a more extensive and diverse dataset that includes a wider variety of food ingredients. Nevertheless, our recommendation system is fully functional, providing users with multiple recipe suggestions based on the recognized ingredient. Moreover, it offers options for refining the recipe recommendations, as mentioned in section 4.3. Therefore, we highly recommend further enhancing our research by incorporating a new and more comprehensive dataset as well as expanding the capability to recognize a broader range of ingredients specified by the user. This will significantly contribute to the overall completeness and robustness of our work.
